# Physics-Aware Variational Autoencoder for Urban Travel Demand Calibration

**Defu Cao**                                               *defucao@usc.edu*
*University of Southern California*

**Sam Griesemer**                                          *samgriesemer@usc.edu*
*University of Southern California*

**Zijun Cui**[†]                                           *cuizijun@msu.edu*
*University of Southern California*
*Michigan State University*

**Carolina Osorio**                                        *osorioc@google.com*
*Google Research*
*HEC Montréal*

**Yan Liu**                                                *yanliu.cs@usc.edu*
*University of Southern California*

**Reviewed on OpenReview:** *https://openreview.net/forum?id=r5oS1XXbT3*

## Abstract

Urban mobility digital twins are revolutionizing how cities manage increasingly complex transportation systems, enabling real-time optimization across multiple stakeholders, services, and dynamic operations. Central to these digital twins is the origin-destination (OD) calibration problem—estimating travel demand patterns that produce realistic traffic simulations matching observed conditions. However, existing calibration methods face critical limitations: they require a prohibitively large number of expensive simulation runs and struggle with high-dimensional city-scale networks. To mitigate these issues, we introduce ControlVAE, a novel physics-informed neural network approach for sample-efficient OD calibration. Our method leverages traffic flow patterns, embedded in an auxiliary differentiable physics model, to directly calibrate an interpretable neural representation of the OD matrix from observed data. Specifically, we develop a conditional variational autoencoder framework with a controllable cross-attention mechanism that incorporates this traffic simulation model via differentiable physics knowledge. Across the evaluated benchmarks, ControlVAE improves calibration quality over SPSA-family and neural baselines while reducing expensive SUMO usage, especially in limited-budget regimes.

## 1 Introduction

In the face of increasingly complex urban mobility systems, realistic traffic simulations have become essential for effective transportation planning and operations. Cities worldwide are now developing digital twins to navigate a landscape of diverse stakeholders, a multitude of interacting services (from public transit to on-demand ride-sharing), and dynamic operations like surge pricing and traffic-responsive signals. These simulators are critical tools for transportation agencies, enabling them to evaluate infrastructure projects, test new autonomous vehicle policies, and optimize overall system performance. However, for a digital twin to be credible, it requires frequent recalibration as new data becomes available and network configurations

---

[†]Work completed while at USC.

evolve. The primary calibration challenge is origin-destination (OD) demand estimation (Qurashi et al., 2022), the task of identifying the underlying travel patterns that replicate observed traffic conditions. As shown in Figure 1, OD calibration aims to estimate the OD matrix[1] distribution that produces simulated traffic flows matching field data. The accuracy of this matrix is paramount, as it serves as the foundational input that governs all mobility patterns within the simulation, directly impacting the digital twin's reliability.

The origin-destination (OD) calibration problem is a simulation-based statistical inference task (Cranmer et al., 2020; Griesemer et al., 2024) aimed at estimating input OD matrices that reproduce observed traffic conditions. This process faces several challenges, including high computational costs from large-scale simulations (Patterson et al., 2011) and high-dimensional parameter spaces in modern digital twins (Choi et al., 2024). Furthermore, the problem is often underdetermined, meaning multiple OD solutions can adequately explain the observed data. While general-purpose simulation-based optimization (SO) algorithms are a common approach (Spall, 2005), they lack sample efficiency, require a large number of simulation runs to achieve reasonable performance (Tympakianaki, 2018).

Traffic simulators like SUMO (Krajzewicz et al., 2002) are computationally expensive and non-differentiable, with intractable likelihood functions that make traditional calibration approaches resource-intensive. While generative models have seen limited application to OD calibration (Mladenov et al., 2022), approaches like variational Autoencoders (VAEs) (Kingma et al., 2019), Generative Adversarial Networks (GANs) (Goodfellow et al., 2020), and diffusion models (Ho et al., 2020) have demonstrated sample efficiency gains in related domains (Isola et al., 2017; Cao et al., 2024b). Unlike diffusion models' dependency on large datasets and Gaussian assumptions, or GANs' susceptibility to mode collapse and training instability, VAEs offer distinct advantages for OD estimation: stable training in data-scarce scenarios, efficient computing of high-dimensional distributions, and an explicit encoding-decoding structure that can naturally accommodate physical constraints. Given the acknowledged lack of efficient simulation-based optimization algorithms (Tympakianaki, 2018), we bridge data-driven and physics-driven approaches (Takeishi & Kalousis, 2021) by incorporating traffic flow dynamics directly into the VAE latent space, reducing dependency on expensive simulator evaluations while maintaining physical realism.

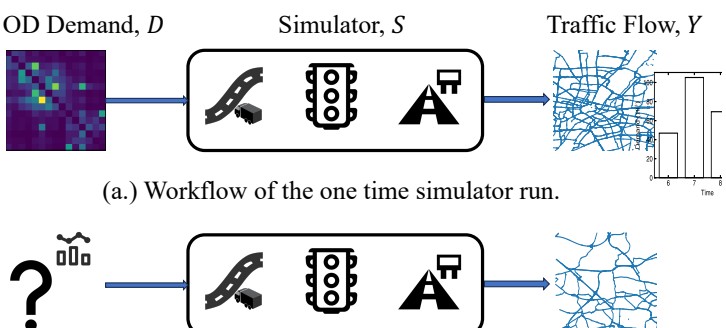

(a.) Workflow of the one time simulator run.

(b.) Origin-destination calibration: identify a desired OD matrix

Figure 1: OD calibration: for a given traffic flow ($Y$), OD calibration aims to identify the desired OD matrix ($D$) distribution that can yield such an observation via a simulator ($S$).

In this paper, we first adopt a Bayesian formulation of the OD calibration problem using a simulation-based framework to train a density estimation model. After that, our proposed method, ControlVAE, yields a probability distribution of OD demands, and we leverage the encoder-decoder VAE architecture as our base generative model. We additionally integrate information from an auxiliary physics traffic model (Arora et al., 2021) into VAE's latent space, giving rise to the proposed physics-aware neural pipeline for the OD calibration task. The physics traffic model takes into account both the potential demand for the OD pair associated with the route[2] and the likelihood that travelers will choose that specific route over other available options. It provides a simplified but computationally efficient description of the mapping of ODs to traffic performance metrics, such as vehicular flows and speeds. The incorporation of physical knowledge is controlled, allowing direct utilization of physical biases during training when sufficient information exists in the representation space, eliminating the need for additional simulation runs. Then, the cross-attention mechanism is intro-

---

[1]An OD matrix tabulates trip volumes between origin-destination zone pairs across the transportation network.

[2]A route is a specific path that connects the starting point to the endpoint. For example, given an OD pair (A to B), a route is represented by the sequence of nodes that a traveler would pass through.

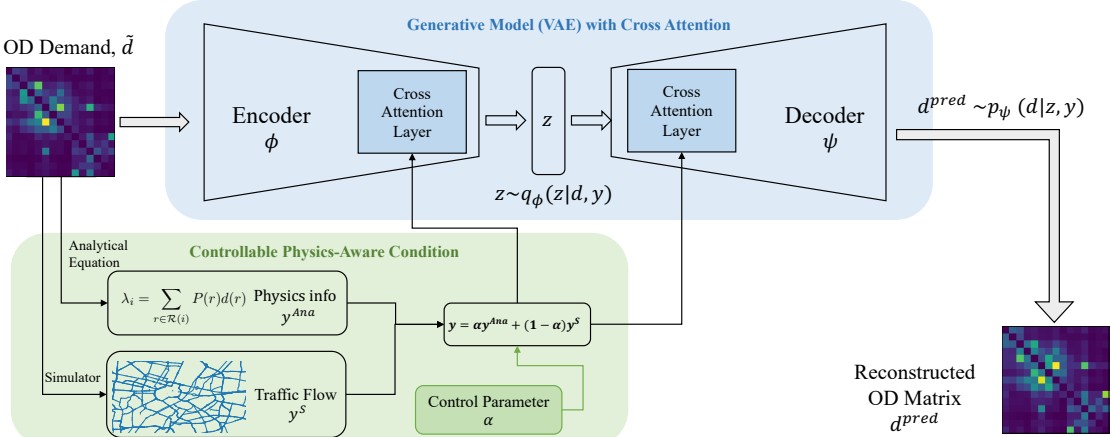

Figure 2: Overview of ControlVAE, the proposed physics-informed generative model. For readability, this schematic focuses on the main conditioning and decoding path; the conditional prior network $p_\eta(\mathbf{z}|y)$ used at inference is described explicitly in the method text and appendix.

duced to discover cross-model interactions and fuse complementary information by adapting the traffic flow's modality to the OD pairs' modality. This provides a flexible way to incorporate different types of knowledge into the OD calibration process. During training, we combine simulator flow and analytical flow into the condition $y$, and jointly train the encoder $q_\phi(\mathbf{z}|d, y)$, conditional prior $p_\eta(\mathbf{z}|y)$, and decoder $p_\psi(d|\mathbf{z}, y)$. During inference, given an observed traffic flow $y_o$, we sample $\mathbf{z} \sim p_\eta(\mathbf{z}|y_o)$ and decode $d \sim p_\psi(d|\mathbf{z}, y_o)$.

In summary, our contributions are three-fold:

- We develop ControlVAE, a novel physics-aware generative framework that reformulates realistic, high-dimensional OD calibration as a Bayesian inference problem. Unlike existing point-estimation methods, our approach learns a posterior distribution over OD matrices by injecting physics constraints into VAE, enabling uncertainty quantification and multiple plausible solutions for underdetermined calibration tasks.

- We design a controllable cross-attention mechanism that seamlessly integrates analytical physical-aware traffic flow equations into the VAE's latent space. This architecture introduces a control parameter $\alpha$ that adaptively balances physics-based guidance with data-driven learning, allowing practitioners to adjust the influence of domain knowledge based on data availability and quality.

- Across the evaluated benchmarks, ControlVAE improves calibration quality over SPSA-family and neural baselines while reducing expensive SUMO usage, with the strongest gains appearing in limited-budget regimes. We therefore frame the main benefit of the physics-aware design as improved sample efficiency and robustness under constrained simulator budgets, rather than a single uniform percentage improvement across all settings.

## 2   Related Work

Previous works primarily approach OD calibration using general-purpose simulation-based optimization (SO) algorithms such as Simultaneous Perturbation Stochastic Approximation (SPSA) methods (Ben-Akiva et al., 2012) and genetic algorithms (Vaze et al., 2009). These general-purpose SO algorithms usually require numerous simulation evaluations, which can be computationally costly. To tackle this issue, recent extensions of SPSA have been proposed (Cipriani et al., 2011). Metamodels have been considered to reduce the high demand for function evaluations (Vishnoi et al., 2023). In particular, the meta-model is a low-resolution

traffic network model that provides a compute-efficient approximation of the simulation. More specifically, we use a model that is formulated as a system of linear equations. Hence it can be solved efficiently for large-scale networks. It has been successfully formulated for high-dimensional OD calibration problems (Osorio, 2019). Nevertheless, meta-models have not yet been explored in the machine-learning community as an effective inductive bias. The urban mobility community has recently called for advances in sample-efficient learning and uncertainty quantification for OD calibration (Choi et al., 2024). While they propose Bayesian optimization with physics-informed kernels, our approach goes further by integrating physics constraints directly into a deep generative model architecture, enabling more expressive posterior distributions and better scalability to high-dimensional problems.

Generative models have gained popularity due to their ability to generate new samples from the training data distribution (Kingma et al., 2019). By generating new samples from the latent space, these models can be used to augment existing datasets, which can improve sample efficiency (Saha et al., 2022). They have a wide range of applications in various fields, such as computer vision (Karras et al., 2019), natural language processing (Bowman et al., 2015), medical imaging (Nie et al., 2018), and time series modeling (Cao et al., 2024a; 2021b; 2025; 2021a). More recently, the diffusion probabilistic model ("diffusion model" in short) has become increasingly popular (Sohl-Dickstein et al., 2015). A diffusion model is a probabilistic model that learns to transform an initial noise distribution into a data distribution, generating samples that match the characteristics of the training data without a change of the dimension (Ho et al., 2020). Generative models, given their ability to handle high-dimensional data, have been widely used in simulation-based inference (Cranmer et al., 2020) and for latent-variable inference from irregularly-sampled time series (Cao et al., 2023).

To further improve the accuracy and reliability of generative models, particularly when the data are limited or noisy, prior knowledge, as inductive bias, is explored and integrated into generative models (Meng et al., 2022). For example, physics knowledge is incorporated into the diffusion models (Shu et al., 2023; Cao et al., 2026). Physics knowledge has been incorporated into generative models widely through the use of physics-based constraints, derived from physical laws or principles and are used to guide the generation process to produce more physically realistic samples (Yu et al., 2021). For example, in the field of fluid dynamics, generative models can be constrained by the Navier-Stokes equations, which govern the behavior of fluids. By incorporating these equations as constraints, the generative model can produce more realistic simulations of fluid dynamics (Raissi et al., 2019). Beyond constraint-based formulations, neural operators have also been developed to directly learn the solution operators of (coupled) partial differential equations, offering differentiable surrogates for the underlying physical systems (Xiao et al., 2023). Compared with prior physics-informed VAEs and prior generative OD calibration work (Mladenov et al., 2022), our method uses an analytical traffic meta-model as cheap differentiable guidance inside a posterior estimator through controllable cross-attention. Recent 2025 work has also emphasized metropolitan-scale calibration and scalable analytical surrogates for OD estimation (Alanqary et al., 2025; Zhang & Osorio, 2025; Zhang et al., 2025). These studies are complementary to our detector-count-conditioned calibration setting and further motivate scalable hybrid approaches.

To the best of our knowledge, the power of generative models has rarely been explored for addressing the challenges associated with OD calibration, such as high-dimensionality and non-convexity. We acknowledge that the mapping from OD to traffic flows is nonlinear and intricately affected by congestion. In this work, we propose to utilize a deep generative modeling framework along with physics knowledge injected for OD calibration. While recent simulation-based inference methods like ASNPE (Griesemer et al., 2024) improve sample efficiency through active learning, they still require more than 100 simulator calls per calibration instance and treat traffic dynamics as a black box. ASNPE is therefore a relevant SBI comparator for this problem, but we do not include an ASNPE implementation in the current experiments; accordingly, our empirical claims are limited to the evaluated transportation and neural baselines.

## 3   Proposed Method

**Background.**   OD calibration is about figuring out the number of trips between different starting points (Origins) and destinations (Destinations) in a transportation system (i.e., OD matrix). The goal is to

calibrate the distribution of the OD matrix so that its simulated traffic can accurately represent the real traffic conditions we see on the roads. For example, consider a study area with 73 zones. The OD matrix would have 73 rows and 73 columns, with each cell representing trips from an origin zone $i$ to a destination zone $j$. The observed traffic count[3] data could consist of volume measurements for $n$ road segments or zone-to-zone pairs of interest. Through OD calibration, the hypothetical OD matrix is refined to generate simulated traffic volumes that align with these $n$ observed counts.

**Overview.** As shown in Figure 2, our proposed approach leverages the generative ability of the VAE and utilizes the underlying physical mechanisms of traffic networks to yield a probability distribution over possible OD solutions in a simulation-sample efficient manner. Specifically, we introduce the physics traffic model which comes from the meta-model simulation-based optimization (SO) approach (Arora et al., 2021) and takes into account both the potential demand for the OD pair and the likelihood for specific routes. This physics traffic model is seamlessly incorporated into the VAE architecture, offering a synergy between detailed link interactions and efficient traffic flow decomposition. Subsequently, a cross-attention mechanism is employed to enhance the model's ability to capture and merge interactions between traffic flow dynamics and OD pair distributions.

### 3.1 Problem Formulation

During a time interval of interest, we consider a single OD matrix $d = \{d_{pair}\}_{pair \in \mathcal{PA}} \in \mathbb{R}^Z$, where $d_z$ represents the expected travel demand for OD pair and $\mathcal{PA}$ is the set of OD pairs. The goal is to match the simulated network performance to real-world measurements. The performance measures used for comparison are typically the expected link counts or speeds. Conventionally, this problem is formulated as a simulation-based optimization problem using a simulator $\mathcal{S}(\cdot; u)$, where $u$ is a vector of simulation variables. The objective consists of the discrepancy between actual link traffic conditions $\hat{y}$ and their simulated counterparts from the simulator, i.e., $\mathcal{S}(d; u)$. Please refer to Appendix C for the conventional problem definition and the difference between OD matrix estimation and OD calibration.

To help quantify underdetermination and enable greater flexibility when choosing OD point estimates, we formulate the calibration problem under the Bayesian paradigm. Instead of obtaining a single calibrated point estimate $d^*$ for an observed traffic flow $\hat{y}$, we now seek a posterior $p(d|\hat{y})$. During testing, given a new observation $y_o$, samples $d \sim p(d|y_o)$ are OD matrices likely to have yielded the observation $y_o$ under the likelihood $p_{\mathcal{S}}(y_o|d)$ implicitly defined by the traffic simulator $\mathcal{S}$. The approximate posterior helps capture intrinsic uncertainty in the calibration problem, leaves open the possibility of different point estimates, and can be used downstream as an informative prior.

More formally, we seek the posterior as seen in the standard setup for Bayesian inference:

$$p(d|y, d^{\mathrm{prior}}) = \frac{p(y|d)p(d|d^{\mathrm{prior}})}{p(y|d^{\mathrm{prior}})} = \frac{p(y|d)p(d|d^{\mathrm{prior}})}{\int p(y|d)p(d|d^{\mathrm{prior}})dd} \tag{1}$$

where $p(d|d^{\mathrm{prior}})$ defines the prior distribution, conditional on the provided noisy OD $d^{\mathrm{prior}}$, which is typically estimated by expert knowledge. This achieves parity with existing approaches, where $d^{\mathrm{prior}}$ is treated as a noisy starting point. For the likelihood, we view the traffic simulator $\mathcal{S}$ as implicitly defining $p(y|d)$:

$$p(y|d) = \int p_{\mathcal{S}}(y, h|d)dh = \int p_{\mathcal{S}}(y, u|d)du, \tag{2}$$

i.e., marginalizing over the possible latent representation $h$ in the simulator's latent space. This integral is intractable for sufficiently complex simulators, necessitating the use of likelihood-free inference methods. Here we turn to recent advances in simulation-based inference, where it is common to train a generative model to directly estimate the posterior (Greenberg et al., 2019; Papamakarios & Murray, 2016; Papamakarios et al., 2017; Bishop, 1994). In our paper, we propose the use of a conditional VAE capable of handling relevant external physics knowledge. When these auxiliary physics models match the underlying simulator

---

[3]Traffic count is the quantification of the number of vehicles passing a specific location, typically measured in vehicles per hour (VPH) or vehicles per day (VPD).

dynamics, they serve as cheap guidance that can reduce erratic behavior during the training stage and improve robustness (Takeishi & Kalousis, 2021). Below we introduce how to combine distinguished physical information with trainable components of the VAE via a cross-attention mechanism.

## 3.2 Physics Traffic Model

In this work, we seek to encode physics mechanisms that approximate the mapping between OD matrix and the simulation-based performance measures (e.g., link flows, $\lambda$ in this section). The proposed physics traffic model explains how alterations in demand can affect the selection of routes and, consequently, influence the geographical dispersion of traffic congestion across the traffic network. In particular, the linear approximation of a simulator $\mathcal{S}$ on link $i$ is defined as (Arora et al., 2021):

$$\lambda_i = \sum_{r \in \mathcal{R}(i)} P(r)f(r), \tag{3}$$

where $\mathcal{R}(i)$ denotes the set of routes that travel through link $i$, $f(r)$ denotes the OD pair of route $r$ and $P(r)$ denotes the probability of choosing route $r$. The right hand side of Equation (3) represents the total expected demand for link $i$, which is calculated as the sum of the expected demand for all routes that include link $i$. That is, the expected route demand is the predicted number of travelers who will use a particular route based on their travel behavior and preferences. In addition, the choice probability of a route $r$, i.e., $P(r)$ is a multinomial logit model with a utility function that depends on the route's travel time $t_r$, which can be computed as:

$$P(r) = \frac{\exp(\theta t_r)}{\sum_{j \in \mathcal{R}(r)} \exp(\theta t_j)}, \tag{4}$$

where $t_j$ denotes the travel time of a given route $j$, typically from the expert knowledge, $\mathcal{R}(r)$ denotes the set of routes that share the same OD pair as route $r$, and $\theta$ is a travel time scalar parameter. This physics traffic model can allow for an analytical description of impacts on the traffic network of traffic assignments, and the solution can be differentiable. The physics model (Eq. equation 3) is defined as a system of linear equations. It maps the unknown OD $d$ to the traffic flows $\lambda$. The injection of the physics traffic model will be introduced in the following section.

## 3.3 Controllable Physics-informed VAE

**Conditional variational autoencoder (CVAE).** In a Conditional Variational Autoencoder (CVAE), given OD matrix data $d$ and a condition $y$, the latent variable $\mathbf{z}$ is learned as a representation of $d$ informed by $y$. The architecture consists of an encoder, $\phi$, which learns to approximate $q_\phi(\mathbf{z}|d, y)$, a conditional prior, $\eta$, which learns $p_\eta(\mathbf{z}|y)$, and a decoder, $\psi$, which aims to reconstruct $p_\psi(d|\mathbf{z}, y)$. Here, we adopt the CVAE as our backbone for modeling the posterior distribution of OD matrix $d$. Traffic flows will be used as condition variables. Particularly, we sample a set of ODs $\{\tilde{d}_m\}_{m=1}^M$ from $p(d|d^{\mathrm{prior}})$, where $m$ indices samples with the total number of samples being $M$. For each sampled OD $\tilde{d}_m$, we obtain a set of corresponding simulated traffic flows $\{\tilde{y}_m^s\}$ as well as the physics information $\{y_m^{Ana}\}$.

**Cross-Attention Fusion.** The traffic flow and OD pairs are under the same physics mechanisms but have different modalities. Specifically, the traffic flows are first-order information from the observed environments, and the OD pairs can be represented by assigning traffic flows to the target OD with a specific probability. To effectively take those two different inputs into the conditional latent space, cross-attention is considered for the fusion to learn the relationships between two different modalities (Tsai et al., 2019). For the multi-head cross-attention (MHCA) with $H$ head, the attention operation of sub-space $h \in \{0, \ldots, H-1\}$ can be computed as:

$$\mathrm{Att}_h(Q_h, K_h, V_h) = \mathrm{Softmax}\left(\frac{Q_h K_h^T}{\sqrt{d_k}}\right) V_h,$$
$$\text{where } Q_h = W_h^Q f(d); K_h = W_h^K g(y); V_h = W_h^V g(y).$$

The learnable projection parameters are query weights $W_h^Q \in \mathbb{R}^{d_{\text{model}} \times d_q}$, key weights $W_h^K \in \mathbb{R}^{d_{\text{model}} \times d_k}$, and value weights $W_h^V \in \mathbb{R}^{d_{\text{model}} \times d_v}$. $d_*$ are the dimensions of corresponding variables. $f(\cdot)$ and $g(\cdot)$ denote the two fully connected layers with trainable parameters for the representation of input $d$ and conditional information $y$, respectively.

**Controllable Physics Injection.** Recall the physics traffic model introduced in the Equation (3), which can be rewritten as $\lambda = Pd$. The matrix $P$ is the probability assignment matrix which maps the OD demand $d$ to traffic flow $\lambda$. It is constructed from route membership and route-choice probabilities in the analytical model and is not learned end-to-end. The traffic flow given by the physics traffic model is then fed into a fully connected layer, $y^{Ana} = l(\lambda)$, where $l(\cdot)$ is a fully connected neural network. The outputs of attention operations are concatenated to capture richer interpretations of inputs and passed through a linear layer to obtain the approximated latent space $\boldsymbol{z}$. Note that the surrogate model usually offers a lower degree of realism compared to the simulator; rigidly imposing its physical constraints would not yield additional benefits. Instead, this surrogate model excels in its differentiability and its capacity for instant evaluation, qualities that make it an approximative tool for introducing a computationally efficient inductive bias. This strategic use of the surrogate model markedly heightens both the sample and computational efficiency of our framework. In that way, we can enhance the latent features $\boldsymbol{z}$ of the OD pairs by using an attention mask derived from the other traffic flow modality. The posterior $q_\phi(\boldsymbol{z}|d, y)$ is modeled as the Gaussian distribution $\boldsymbol{N} \sim (\mu_z, diag(\sigma_z^2))$. This diagonal Gaussian assumption is imposed only in latent space $\boldsymbol{z}$, not directly on OD entries. Dependencies among OD pairs are captured by the shared nonlinear decoder and conditioning pathway.

To strike a balance between physics-based and data-driven models and prevent detrimental effects during the learning process, we introduce a control parameter $\alpha$, which is a scalar with a value between 0 and 1. It controls the importance of the physics traffic model $y^{Ana}$ and simulation $y^s$ for each OD matrix, i.e.,

$$y = \alpha y^{Ana} + (1 - \alpha) y^s. \tag{5}$$

In practice, we tune the hyperparameter $\alpha$ using a grid search approach to estimate an approximate OD value. It can also be set by users at inference to adjust the rule strength, with the availability of the sample (Seo et al., 2021). Subsequently, the traffic flow data, in conjunction with both physical constraints as well as information from the simulator, are utilized as keys and values in a cross-attention mechanism within the VAE for conditioning on the actual origin-destination distribution. This strategic use of the surrogate model markedly heightens both the sample and computational efficiency of our framework. By doing so, it considerably diminishes the model's dependency on voluminous simulation data, often a limiting factor in such research. The analytical equations serve as the corresponding condition information, which is pivotal for training our generative model without incurring substantial computational overhead. In essence, our approach smartly integrates domain knowledge to guide the learning process effectively, thereby achieving enhanced efficiency and robustness in our solutions.

### 3.4 Training Objective

In practice, the training objective is comprised of two components. One is the variational lower bound of CVAE:

$$\begin{aligned} L_{\text{CVAE}} = &-\mathbb{E}_{\boldsymbol{z} \sim q_\phi(\boldsymbol{z}|d,y)}[\log p_\psi(d|y, \boldsymbol{z})] \\ &+ D_{KL}(q_\phi(\boldsymbol{z}|d, y)||p_\eta(\boldsymbol{z}|y)), \end{aligned} \tag{6}$$

where $D_{\text{KL}}$ represents the Kullback-Leibler (KL) divergence between two distributions. Optimization over each of the involved neural networks is performed via amortized variational inference, including the recognition network $q_\phi(\boldsymbol{z}|d, y)$, generation network $p_\psi(d|y, \boldsymbol{z})$, and conditional prior network $p_\eta(\boldsymbol{z}|y)$. During training, the combined condition $y$ from Eq. 5 is used by all three components. The second component of the training loss is the mean squared error (MSE) between the $y^{Ana}$ and the traffic flow $y^s$ generated from the simulator:

$$L_{\text{MSE}} = ||y^{Ana} - y^s||_2^2. \tag{7}$$

$L_{\mathrm{MSE}}$ is a regularizer designed for aligning the physics information with the simulator's behavior. Together, the total loss includes both of these terms as follows: $L = L_{\mathrm{CVAE}} + \gamma L_{\mathrm{MSE}}$, where $\gamma$ controls the strength of the physics-alignment regularizer. In our experiments, $\gamma$ is tuned per scenario (Appendix E). The SUMO simulator is fixed and non-differentiable in our setup; gradients flow through the analytical branch $y^{Ana}$ and the neural networks, but not through the simulator outputs $y^s$.

During the training process, the CVAE's generation network $p_\psi(d|y, \boldsymbol{z})$ learns a distribution approximating the posterior in Equation (1) (the training data are generated according to $(y, d) \sim p(y|d)p(d) \propto p(d|y)$). During the inference stage, samples can be drawn from this distribution by first sampling $\boldsymbol{z} \sim p_\eta(\boldsymbol{z}|y)$, then drawing $d \sim p_\psi(d|y, \boldsymbol{z})$ from the learned generation network. Accordingly, our method is a conditional posterior estimator $p(d|y)$ rather than an unconditional generator of OD matrices. When conditioning on an observed traffic flow of interest $y_o$, ODs $d$ drawn under this sampling procedure can be viewed as estimates for the calibration problem that include both assumptions from the prior and information gained from simulation evaluations.

## 4 Experiments

In the explored experimental settings, our goal is to verify that the developed algorithm is computationally efficient and can find high-quality solutions within a limited number of simulation runs. In practice, calibration algorithms operate under tight computational constraints where limited numbers of simulation function evaluations can be performed. The design of sample-efficient algorithms in this space is thus directly in line with the needs of practitioners.

### 4.1 Experiment Settings

We conducted our experiments on the large-scale Munich regional network (Qurashi et al., 2022) and the widely used Sioux Falls benchmark. The Munich network is a high-dimensional road network with 5,329 origin-destination (OD) pairs, and 507 detector locations are incorporated in this case study. Note that it is a highly underdetermined system with 5,329 unknowns for each of the time intervals and only 507 traffic measurements. The Sioux Falls network, configured with 9 OD zones and 39 sensors, serves as an ideal benchmark network for testing OD estimation and calibration methods because it provides a balanced combination of complexity and manageability. For Munich, the target OD scenarios are generated by synthetic perturbations of a historical OD estimate following prior benchmarking protocols. Please refer to Appendix D for detailed information on the SUMO simulator and Appendix E for our experiment settings on the definition of Set I and Set II.

### 4.2 Quantitative Results with Baselines

We evaluate the effectiveness of the proposed solution by comparing ControlVAE against state-of-the-art (SOTA) methods from both transportation and machine learning domains. From the transportation domain, we compare with Simultaneous Perturbation Stochastic Approximation (SPSA) (Spall, 1992) and its principal component-based variant, PC-SPSA (Qurashi et al., 2022). SPSA is widely adopted for travel demand calibration, while PC-SPSA enhances SPSA by reducing the optimization search space to a lower dimension. These conventional optimization-based methods operate serially without parallel computing capabilities and do not utilize neural networks, referring to Appendix F. To comprehensively assess our neural approach, we establish several neural baselines. We first consider the vanilla conditional variational autoencoder (CVAE). We then evaluate two variants of our approach: 1) CVAE-catt, which incorporates only the cross-attention module without physics equations, and 2) CVAE-phy, which integrates physics equations directly into CVAE without the cross-attention module. Given the scarcity of practical models for this problem, these variants serve both as baselines and as components of our ablation study. Additionally, we report the RMSN between the traffic flow generated by the perturbed OD demand and the real traffic flow (denoted as "Original"). Table 1 reports RMSN on traffic counts rather than on raw OD entries. For neural posterior models, we draw ten OD samples from the learned posterior, simulate the corresponding traffic counts, average those ten

Table 1: ControlVAE's results with the comparison to Transportation Baselines (SPSA, PC-SPSA) and Neural Baselines (CVAE, CVAE-catt, CVAE-phy) on the normalized root mean squared error (RMSN (%)). For neural posterior models, we draw ten OD samples from the learned posterior, simulate the corresponding traffic counts, average the ten simulated traffic-count vectors into a deterministic prediction, and compute RMSN against the ground-truth traffic counts. Mean±std are reported over repeated runs. [†]PC-SPSA is omitted on Sioux Falls because our current PCA-based initialization pipeline was prepared only for the Munich benchmark.

|  |  | Munich 5-6 | Munich 6-7 | Munich 7-8 | Munich 8-9 | Munich 9-10 | SiouxFalls |
|---|---|---|---|---|---|---|---|
| Set I | Original | 97.08±1.23 | 52.20±0.72 | 36.40±1.63 | 49.76±0.80 | 43.92±0.56 | 30.2±0.78 |
|  | SPSA | 24.67±1.81 | 24.59±2.41 | 21.24±1.33 | 47.06±0.64 | 18.40±0.44 | 28.1±0.02 |
|  | PC-SPSA | **15.40**±2.71 | 35.05±0.44 | 22.64±2.61 | 28.36±4.66 | 21.94±0.51 | —[†] |
|  | CVAE | 22.00±1.59 | 22.98±2.15 | 19.28±2.81 | 23.31±2.81 | 30.56±1.09 | 26.5±0.01 |
|  | CVAE-catt | 18.45±0.61 | 22.41±2.00 | 19.33±1.70 | 20.96±0.82 | 16.34±1.17 | 27.9±0.02 |
|  | CVAE-phy | 21.43±2.85 | 22.04±2.48 | 20.78±1.35 | 23.92±2.37 | 17.48±1.30 | 26.8±0.02 |
|  | **ControlVAE** | 17.40±0.87 | **22.02**±1.45 | **17.55**±1.29 | **19.89**±1.84 | **16.28**±1.22 | **23.7**±0.01 |
| Set II | Original | 97.20±1.62 | 87.52±1.62 | 101.3±3.15 | 70.21±0.64 | 80.70±0.58 | 40.1±0.53 |
|  | SPSA | 18.00±1.12 | 43.10±0.24 | 55.89±2.31 | 50.04± 0.61 | 36.13±0.43 | 30.4±0.07 |
|  | PC-SPSA | 15.03±0.71 | 35.66±0.42 | 23.46±3.23 | 28.79±0.51 | 22.31±0.51 | —[†] |
|  | CVAE | 46.23±0.91 | 24.57±1.90 | 26.28±1.60 | 27.09±1.69 | 18.25±0.67 | 27.1±0.04 |
|  | CVAE-catt | 16.43±0.79 | 30.72±1.76 | 18.47±1.75 | 21.60±1.55 | 19.42±1.15 | 27.5±0.03 |
|  | CVAE-phy | 15.73±1.00 | 22.75±1.71 | 21.76±3.08 | 28.63±1.86 | 17.93±1.04 | 29.4±0.04 |
|  | **ControlVAE** | **14.89**±0.56 | **21.74**±1.59 | **18.32**±1.83 | **21.02**±1.84 | **16.38**±1.02 | **22.7**±0.02 |

simulated traffic-count vectors into a deterministic prediction, and compute RMSN against the ground-truth traffic counts. For SPSA and PC-SPSA, mean±std are reported over ten converged runs.

Our comprehensive evaluation compares the normalized root mean squared error (RMSN or NRMSE) on traffic counts. As shown in Table 1, ControlVAE consistently improves over SPSA and performs strongly among the evaluated neural baselines. Additional low-budget analysis is provided in Appendix G.1.

### 4.3 Qualitative Results on the Generated Case

**Generated OD Distribution.** At inference time, when presented with a new OD demand estimation and traffic count observation, the model employs an attention mechanism to highlight the region of the latent space corresponding to the true underlying OD demand, enabling the generation of a fitted probability distribution centered around the ground truth. Figure 3 (a) - (c) and Figure 5 in the Appendix demonstrate the effectiveness of our model on large-scale datasets to assign high probability to the true OD solution.

**Traffic Flow Calibration.** We compare the observed and simulated traffic counts for all detector locations and plot the traffic counts in the pre-calibration data against the average of ten simulation samples from the OD distribution obtained after calibration using Control VAE in Figure 3 (d) - (e). For all intervals, the proposed approach identifies solutions that replicate well the true counts (i.e.,the points lie along the 45-degree line). For the morning peak period of 8:00 am-9:00 am, when the traffic counts of the initial OD are furthest from their true value, the proposed method identifies ODs that yield a great fit to the real data which holds for all detector locations.

**Convergence Speed.** We compare the convergence curves for the best RMSN achieved by SPSA, PC-SPSA, and our proposed ControlVAE method. For fair comparison, we plot the results against the number of simulator executions rather than wall-clock runtime. For display, SPSA and PC-SPSA are plotted from the 1st iteration, while ControlVAE is plotted from the 10th SUMO run. As shown in Figure 3 (f), ControlVAE rapidly converges once the target result is obtained using just 10 SUMO runs. Considering the parallelizable data generation, this demonstrates the data efficiency of ControlVAE. Figure 3 (f) reports simulator-call efficiency; the corresponding wall-clock discussion is provided in Appendix E. By leveraging both simulation

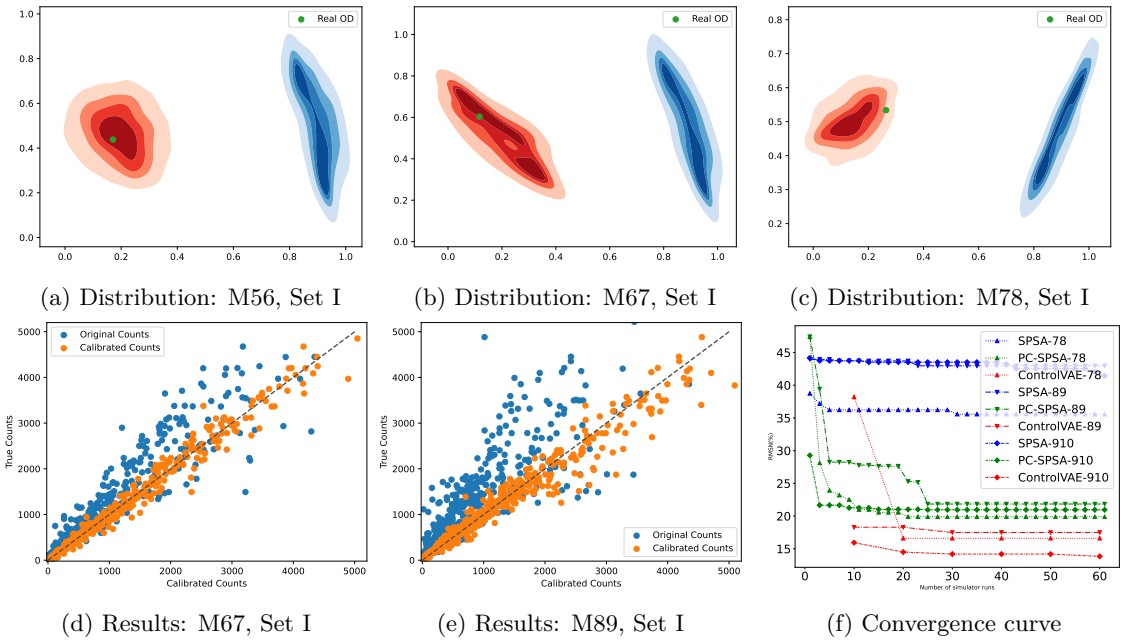

Figure 3: Calibration Distribution and Results on OD demand. For subplot (a) - (c), the blue cluster is the prior distribution of input OD and the red cluster is the calibrated OD distribution condition on observed traffic counts. The green dot refers to the real OD that we aim to identify. For subplot (d) - (e), the orange dots, which are different detector locations, represent the calibrated results (orange dots) given by ControlVAE. Subplot (f) is the convergence curve with different simulator running times. The legend is named after the model-time interval; for example, SPSA-78 means the SPSA method with a time interval from 7 am to 8 am according to the settings in this paper.

and analytical models with attention mechanisms, ControlVAE achieves accurate OD matrix calibration with fewer required simulator evaluations. Please refer to Section G.2 for a detailed analysis on convergence speed.

**Controllability on Physics Awareness.** As detailed in the Section G.1, our further analysis of $y^{Ana}$ introduced in Section 3.3 demonstrates how the controllable integration of physical knowledge substantially reduces simulation computational overhead.

Table 2: Performance Comparison of Advanced Methods

| Time Period | CVAE (SPSA init) | Transformer | Diffusion | ControlVAE |
|---|---|---|---|---|
| Munich 5-6 | 15.97 | 16.71 | 80.50 | **14.89** |
| Munich 6-7 | 25.80 | 23.58 | 86.48 | **21.74** |
| Munich 7-8 | 29.25 | 20.21 | 83.48 | **18.32** |
| Munich 8-9 | 21.54 | 23.32 | 85.75 | **21.02** |
| Munich 9-10 | 19.00 | 16.48 | 82.10 | **16.38** |

## 4.4 Extended Model Comparisons

We conduct comprehensive evaluations by comparing our proposed method against the potential improvement of traditional approaches, i.e., CVAE-initiated SPSA and potential complex models, i.e., Transformers (Ramana et al., 2023) and Diffusion models (Jiang et al., 2024) . As shown in Table 2, while CVAE initialization improves SPSA's performance, it still falls short of our method. The transformer model is competitive, while the diffusion baseline underperforms under our current low-data protocol. We interpret this result as evidence about an off-the-shelf diffusion configuration in the present benchmark setting, not

as a definitive statement that diffusion models are inherently unsuitable for OD calibration. Please refer to Section G.3 for the corresponding discussion.

## 5 Conclusion

In this work, we attempt to address critical drawbacks of commonly employed methods for OD calibration by introducing an approach that combines conditional variational autoencoders with physics-based traffic models. Leveraging deep generative models for this task constitutes a generally under-explored area, and through our problem formulation we aim to underscore its importance and tractability to the machine learning community. The integration of deep learning with domain-specific physical constraints not only advances transportation modeling but also establishes a paradigm for incorporating domain knowledge into machine learning systems, with potential applications across various urban planning tasks. Across the evaluated benchmarks, ControlVAE improves calibration quality over SPSA-family and neural baselines while reducing expensive SUMO usage, especially in limited-budget regimes.

## Acknowledgement

This work is partially supported by the NSF Award #2425919, and NSF Award #2413417. The funding from these sources has been a cornerstone in enabling us to bring our project to fruition. We are also deeply grateful to the anonymous reviewers for their rigorous review process. Their detailed comments and constructive suggestions have significantly contributed to the improvement of this paper.

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

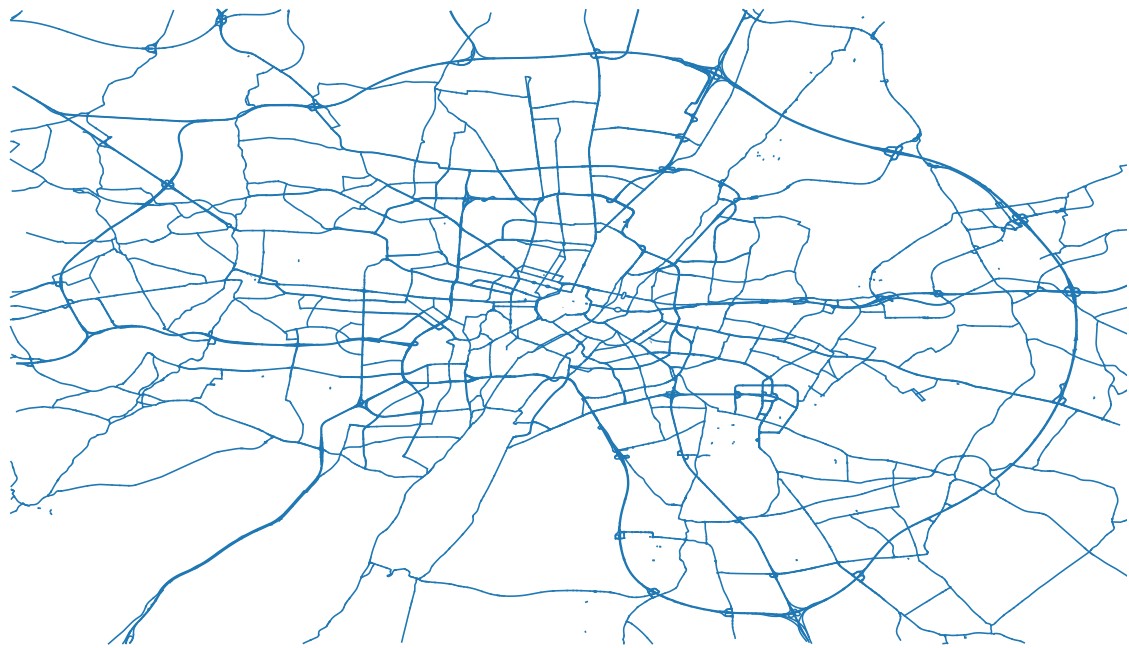

Figure 4: Overview of Munich Traffic Network.

# A    Code Reproducibility

The reproducible code of the paper can be found at the Supplementary Material and is publicly available at `https://github.com/DC-research/ReLATE`.

# B    Broader Impact

**Efforts on transportation networks:** The availability of real-world, metropolitan-scale, road traffic simulators is very limited. To our knowledge, no published work evaluates the work on more than one real-world metropolitan-scale traffic simulation model. Most high-impact works in transportation limit their validation to synthetic networks or small-scale networks. In the paper, we present a large-scale, and challenging, Munich case study. Due to the difficulty of developing models of large-scale networks, there is also a lack of benchmark problems for calibration. There are parallel efforts by co-authors of this paper to open source and develop future benchmarks.

**Additional discussion.** Machine learning methods, particularly deep generative models such as Variational Autoencoders (VAEs), are a flexible way to represent high-dimensional OD distributions without explicitly specifying all nonlinear interactions between OD demand and observed traffic conditions.

**Broader impact:** The application of machine learning to transportation problems such as OD calibration is not only a technical achievement but also a major step towards more sustainable, efficient, and user-friendly urban environments. It paves the way for the development of smart cities where data-driven insights can inform infrastructure development, urban planning, and the deployment of autonomous vehicles. As these models continue to improve and become integrated into real-world applications, their impact on society's mobility and quality of life will be significant and enduring.

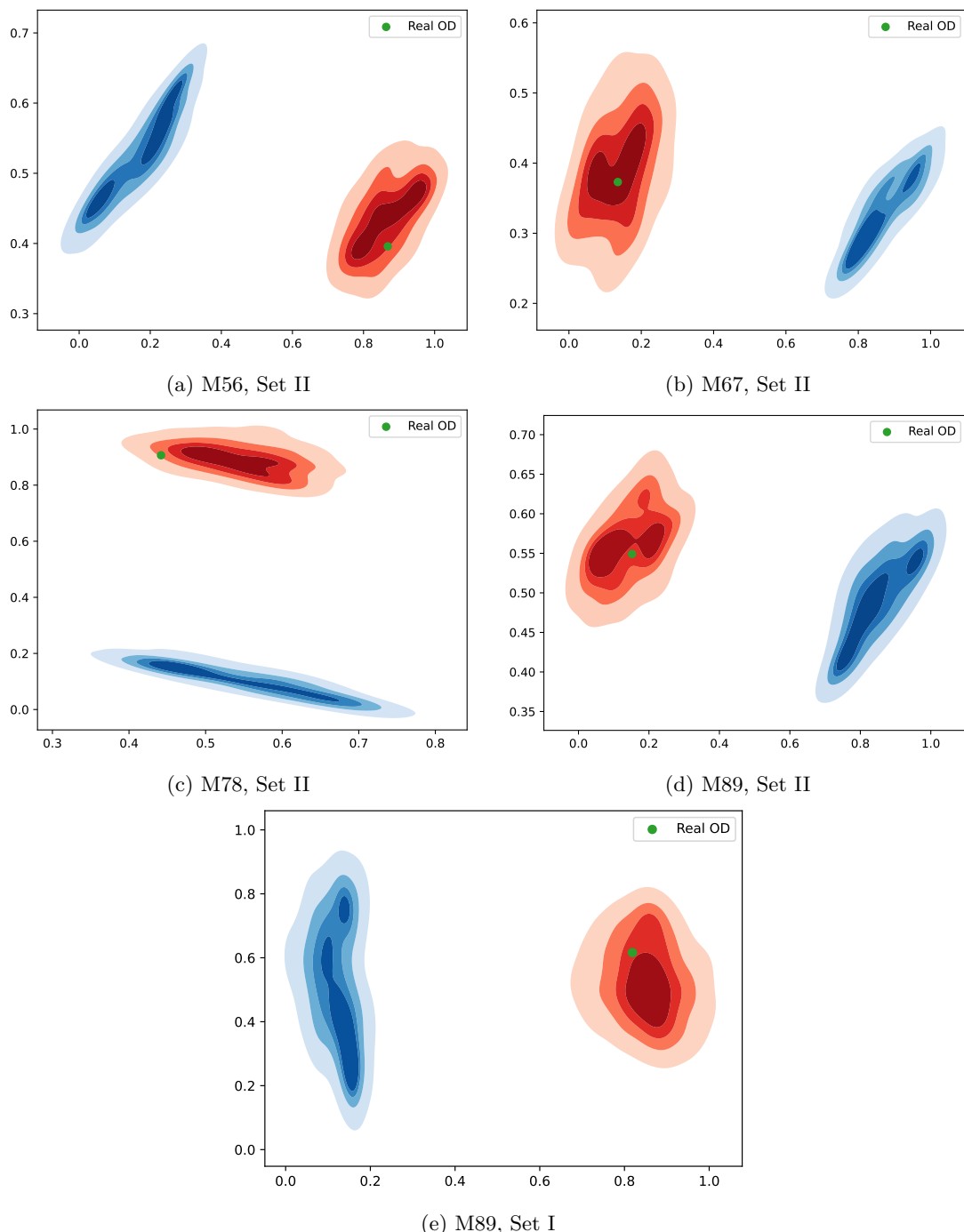

Figure 5: Calibration distribution results on OD demand. The blue cluster is the prior distribution of input OD and the red cluster is the calibrated OD distribution conditioned on observed traffic counts. The green dot refers to the real OD that we aim to identify. These appendix plots further illustrate that the learned posterior concentrates near the target OD while retaining multiple plausible solutions. They should be read as qualitative diagnostics of the conditional posterior for fixed observations, rather than as unconditional samples from a standalone OD prior.

## C Preliminary

### C.1 Conventional Problem Definition

In this section, we provide the traditional problem definition of origin-destination (OD) calibration in the transportation science domain. In this paper, we consider the static OD calibration problem, where the OD matrix is usually determined offline based on historical field measurements. During the time interval of interest, we consider a single OD matrix $d = \{d_z\}_{z \in \mathcal{Z}}$. $d_z$ represents the expected travel demand for OD pair $z$ and $\mathcal{Z}$ is the set of OD pair indices, i.e., $\mathcal{Z} = \{1, 2, \ldots, Z\}$. $Z$ is the total number of OD pairs. This problem aims to calibrate the OD matrix to match the simulated network performance to real-world measurements. The performance measures used for comparison are typically the expected link counts or speeds. Conventionally, this problem is formulated as a simulation-based optimization problem using a simulator $\mathcal{S}(\cdot; u_1, u_2)$. $u_1, u_2$ are vectors of endogenous simulation variables and exogenous simulation parameters, respectively. The objective consists of identifying an OD matrix that leads to simulated traffic metrics that are reflective of traffic conditions observed from the field, e.g.,

$$\min_d f(d) = \frac{1}{|\mathcal{I}|} \sum_{i \in \mathcal{I}} (y_i - \mathbb{E}\left[\mathcal{S}_i\left(d; u_1, u_2\right)\right])^2 + \delta \frac{1}{|\mathcal{Z}|} \sum_{z \in \mathcal{Z}} \left(d_z^{\mathrm{prior}} - d_z\right)^2 \tag{8}$$
$$0 \leqslant d \leqslant d^{\mathrm{max}},$$

where symbols $|\mathcal{I}|$ and $|\mathcal{Z}|$ represent the number of elements in sets $\mathcal{I}$ and $\mathcal{Z}$, respectively. $\mathcal{S}_i$ is the simulated flow on link $i$ and $y_i$ is the average flow on link $i$ estimated from the observed field data. $d_z^{\mathrm{prior}}$ is a prior value for the expected demand for OD pair $z$ and $\delta$ is a weight parameter for prior information. The problem is formulated with bound constraints, which is $0 \leqslant \theta \leqslant \theta^{\mathrm{max}}$ and $d^{\mathrm{max}}$ is an upper bound vector. The primary objective of Eq 8's first summation is to assess the discrepancy between actual link traffic conditions (represented by the terms $y_i$) and their simulated counterparts (represented by the terms $\mathbb{E}\left[\mathcal{S}_i\left(d; u_1, u_2\right)\right]$). In addition, there exists an infinite set of OD vectors, $d$, that can lead to the same simulated traffic conditions. However, some solutions may be more physically plausible than others, as they reflect the underlying land-use patterns of the region, resulting in plausible travel demand patterns for a specific time period. To address this underdetermination issue, the traditional approach is to add a term to the objective function (second term in Eq 8) that encourages the OD solutions to be close to a plausible pre-defined OD matrix. This pre-defined OD matrix is called a prior or seed matrix and is most often estimated from other data sources, such as census data, or obtained through OD estimation of a simpler (e.g., static) traffic model. Note that the second term of the objective function is a regularization term and presents as an analytical function so we do not need to evaluate via simulation. However, the regularization term is a useful tool to impose additional constraints on the problem, particularly in situations where the problem is underdetermined due to the type and sparsity of the measurements available. It is assumed to capture plausible travel demand patterns for the calibration time period under consideration. $\delta$ is a weight factor that determines the trade-off between the distance to field measurements (first summation) and the distance to the prior OD matrix (second summation). Incorporating this prior knowledge into the calibration problem can improve the plausibility of the estimated OD vectors, leading to more accurate simulations of the transportation network.

### C.2 Difference between OD Matrix Estimation and OD Calibration

We distinguish between two problem formulations: (i) general origin-destination (OD) matrix estimation, where OD matrices are inferred for various network analyses. Specifically, OD matrix estimation is the process of estimating the number of trips that are made between origins and destinations in a transportation network; (ii) OD calibration, where inputs such as OD matrices are estimated to calibrate a specific traffic model. Traffic models have numerous demand and supply parameters requiring calibration, and their values should not be interpreted in isolation, especially for calibration problems. Calibrated OD matrices should not be utilized as standalone matrices for planning, as they depend on other model input parameters (Cascetta et al., 2013). While both problem classes can share mathematical formulations, the former class often leverages a traffic model to regularize underdetermined problems by introducing behavioral assumptions or

network knowledge to further constrain the problem. This paper focuses on the second problem class, aiming to calibrate the OD matrix input of a traffic simulator for network planning and operations decisions.

## D  The Simulator – SUMO

Simulation of Urban MObility (SUMO)[4] is an open-source, highly portable, microscopic and continuous multi-modal traffic simulation package designed to handle large networks. It allows for intermodal simulation, including pedestrians and comes with a large set of tools for scenario creation (Lopez et al., 2018). SUMO has a wealth of features that allow for the modeling of intermodal traffic systems, including road vehicles, public transport, and pedestrians. Some of its features include automated driving, vehicle communication, traffic management, microscopic simulation, multimodal traffic, online interaction, network import, demand generation, traffic lights, performance, and portability. To accurately capture real-world dynamics, in this work, the simulations utilize a meticulous mesoscopic modeling approach where each individual trip in the system is modeled as a one-time random route selection by the user based on their personal preferences and the current traffic conditions. A recent paper presented an efficient simulation-based travel demand calibration algorithm for large-scale metropolitan traffic models using SUMO (Arora et al., 2021). The algorithm builds upon recent metamodel methods that tackle the simulation-based problem by solving a sequence of approximate analytical optimization problems, which rely on the use of analytical network models. Besides, SUMO has been used for a wide variety of applications. Some of its application areas include evaluating the performance of traffic lights, vehicle route choice, providing traffic forecasts, supporting simulated in-vehicle telephony behavior for evaluating the performance of GSM-based traffic surveillance, and providing realistic vehicle traces (Shamim Akhter et al., 2020).

## E  Detailed Experimental Settings

We build the synthetic demand scenarios following an established benchmarking framework (Antoniou et al., 2016; Qurashi et al., 2019; Cantelmo et al., 2019) to test the calibration algorithms fairly. Specifically, we first construct the target demand from the previous estimate of OD demand $\hat{d}$ and the ground truth traffic information (counts) obtained from the simulation output. We then synthetically perturb the target demand to generate different scenarios for calibration. Following (Qurashi et al., 2019), two coefficients - reduction ($p$) and randomization ($q$) - are used for perturbation. Varying these two coefficients creates different types of true demands, as in reality. The demand scenario generation process can be described as: $x_c = (p + q \times \delta) \times \hat{d}$, where the random perturbation vector, $\delta$, follows a Gaussian distribution with $\delta \sim N(0, 0.333)$, which is made to to ensure that 99.7% of the random values fall between -1 and 1. This range is selected to maintain the perturbed demand within a reasonable range while still introducing sufficient randomness to mimic real-world variations and $p = 0.7$. In our experiments, we establish two different scenarios with different degrees of randomness. In Set I, we set $q = 0.15$, while in Set II, we set $q = 0.3$. We hypothesize that incorporating increased randomness in the selection of initial points contributes to greater difficulty in addressing the OD calibration problem. By deliberately introducing heightened uncertainty (Set II) in the starting position, the problem becomes more intricate.

Note that we refer to the hour-by-hour settings as "Munich $X$-$Y$" (and abbreviated "M$XY$" in plots below), where the time range is $X$:00am-$Y$:00am. For example, "Munich 6-7" and "M67" are shorthand for the Munich setting from 6:00 am to 7:00 am. To generate the necessary traffic information (traffic counts for the environments, we use the open-source traffic simulator Simulation of Urban MObility (SUMO) (Lopez et al., 2018). Our approach can also be applied on any other traffic simulator. To account for the stochasticity of the traffic simulations, we used outputs averaged over 5 simulation replications.

The availability of real-world, metropolitan-scale road traffic simulators is very limited. To our best knowledge, no published work evaluates the work on more than one real-world metropolitan-scale traffic simulation model. Most high-impact works in transportation limit their validation to synthetic networks. In the paper, we present a large-scale, and challenging, Munich case study. In addition, we carry out validations on an additional network named Sioux Falls. The Sioux Falls network, configured with 9 OD zones and 39 sensors,

---

[4]https://www.eclipse.org/sumo/

serves as an ideal benchmark network for testing OD estimation and calibration methods because it provides a balanced combination of complexity and manageability. Its structure (24 nodes, 76 links) is complex enough to be realistic yet simple enough for efficient computation, while the 39 sensors offer adequate network coverage for data collection. The 9-zone system creates a reasonably sized 9x9 OD matrix with 81 OD pairs, making it perfect for developing and validating new methodologies in transportation studies without the computational burden of larger networks. For clarity, the Munich experiments still rely on synthetically perturbed benchmark targets derived from historical OD demand because full real-world OD ground truth is unavailable.

Regarding the hyperparameters of the proposed model, if not specified, we use the same hyperparameters in all experiments. Specifically, we set the learning rate as 0.001, the epoch as 1000, the dimension of the hidden space as 10, the weight ($\gamma$) of $L_{\mathrm{MSE}}$ as 1 for M56, M67; 0.01 for M78, M89 and 0.001 for M910, respectively. The optimizer Adam (Kingma & Ba, 2014) and SGD (Kiefer & Wolfowitz, 1952) is adopted to train the model. The encoder, decoder, and conditional prior are implemented as fully connected networks, and the representation maps $f(\cdot)$ and $g(\cdot)$ used in the multi-head cross-attention block each contain two fully connected layers as described in Section 3.3. We tune the control parameter $\alpha$ by grid search and use a latent dimension of 10 in all experiments. We treat $\alpha$ and $\gamma$ as the key sensitivity parameters of ControlVAE. All codes are implemented by Python and we use Intel(R) Core(TM) i9-10900X CPU @ 3.70GHz 64GB memory, and NVIDIA GeForce RTX 2080 Ti. With such device, a single SUMO simulation run takes 12 minutes to complete. SPSA requires a minimum of two simulations per iteration with a runtime of over 24 minutes in general. In contrast, neural network-based approaches utilize a parallel data collection procedure from the Munich traffic network, shown in Figure 4, instead of serial iterations. Note that $n$ mentioned in Section G.2 first paragraph can be up to 40 in our paper, which depends on the CPU kernels. Under this setup, 200 SPSA calls correspond to roughly 40 simulator-hours, while 150 ControlVAE calls can be collected in approximately 48 minutes of simulator time under 40-way parallelism, excluding neural-network training overhead. In addition, to facilitate model training generative models, we apply min-max normalization to scale the input data in the range of 0-1, and re-scale to the original distribution with input OD demand during validation.

## F   Detailed Descriptions on Transportation SOTA Method

- SPSA: SPSA (Simultaneous Perturbation Stochastic approximation) is an optimization algorithm for systems with multiple unknown parameters, which can be used for large-scale models and various applications. It can find global minima, like simulated annealing (Van Laarhoven et al., 1987). SPSA works by approximating the gradient using only two measurements of the objective function with gradients, making it scalable for high-dimensional problems (Spall, 1992).

- PC-SPSA: PC-SPSA is proposed to address the issue of SPSA that it often fails to converge reasonably with the increase in problem size and complexity. This is because SPSA searches for the optimal solution in a high-dimensional space without considering the structural relationships among the variables. PC–SPSA combines SPSA with principal components analysis (PCA) to reduce the problem dimensionality and limit the search noise. The PCA captures the structural patterns from historical estimates and projects them onto a lower-dimensional space, where SPSA can perform more efficiently and effectively (Qurashi et al., 2022; 2019).

We implement the SPSA and PC-SPSA according to `https://github.com/LastStriker11/calibration-modeling` and the parameters used in our work are aligned with this repository. In addition, we employ Method 6, titled "Spatial, Temporal, and Day-to-Day Correlation," as delineated in (Qurashi et al., 2022), to systematically generate historical data for subsequent principal component analysis. According to the original research, method 6 constitutes the most robust and optimal solution for the given context. PC-SPSA is not reported on Sioux Falls because our current implementation of the benchmark-specific PCA initialization pipeline was prepared only for the Munich setting; rather than report a mismatched implementation, we omit that entry.

Note that, SPSA and PC-SPSA require a minimum of two simulations per iteration. We allow each of the two baselines to run up to 100 iterations, with multiple simulations and optimization until convergence. In contrast, our approach does not have this limitation, as we can run multiple parallel simulations and collect the generated OD demands and traffic flows as training data. In practice, we allow a maximum of 300 sequential simulation runs on Set I and 150 simulation runs on Set II, with a maximum of 40 parallel simulations simultaneously.

### F.1 Additional Method Clarifications

For each sampled OD matrix $d$ during training, we compute simulator flow $y^s$ and analytical flow $y^{Ana}$, combine them as $y = \alpha y^{Ana} + (1 - \alpha)y^s$, and use this same conditioning variable for the encoder $q_\phi(\mathbf{z}|d, y)$, conditional prior $p_\eta(\mathbf{z}|y)$, and decoder $p_\psi(d|\mathbf{z}, y)$. At inference time, given an observed traffic flow $y_o$, we sample $\mathbf{z} \sim p_\eta(\mathbf{z}|y_o)$ and decode $d \sim p_\psi(d|\mathbf{z}, y_o)$. This appendix clarification is intended to distinguish our conditional posterior estimation setting from unconditional OD generation.

The role of the conditional prior is to provide a test-time latent distribution conditioned only on the observed flow information. During training, the KL term encourages consistency between the encoder posterior $q_\phi(\mathbf{z}|d, y)$ and the conditional prior $p_\eta(\mathbf{z}|y)$; during inference, only $p_\eta(\mathbf{z}|y_o)$ and $p_\psi(d|\mathbf{z}, y_o)$ are needed. In this sense, $p_\eta$ is not a replacement for the simulator, but a learned latent prior adapted to the traffic observation.

The analytical branch is implemented as $\lambda = Pd$ followed by $y^{Ana} = l(\lambda)$. Here, $P$ is fixed by the analytical route-assignment construction and is not optimized end-to-end, whereas the projection network $l(\cdot)$ and the CVAE components are trainable. The simulator output $y^s$ is obtained from SUMO and treated as non-differentiable. Consequently, gradients from $L_{\mathrm{MSE}} = \|y^{Ana} - y^s\|_2^2$ flow through the analytical and neural components, but not through the simulator itself.

The diagonal Gaussian assumption is imposed only on the latent posterior and conditional prior in $\mathbf{z}$-space. It should not be interpreted as an assumption that OD entries are independent after decoding. Correlations among OD pairs are represented through the shared nonlinear decoder $p_\psi(d|\mathbf{z}, y)$ and the common conditioning variable $y$.

Finally, our uncertainty claims should be interpreted in the conditional sense: the model returns multiple plausible OD samples for a fixed observed traffic pattern $y_o$. The appendix distribution plots therefore serve as qualitative posterior diagnostics, while the comparisons among CVAE, CVAE-catt, CVAE-phy, and ControlVAE in the quantitative table should be read as the main ablation of conditioning and physics injection.

We did not fully redraw Figure 2 in the revision because our goal here was to keep the main-paper edits minimal. Instead, we clarified the train/inference distinction directly in the introduction, method section, Figure 2 caption, and this appendix note. In particular, the main path shown in Figure 2 corresponds to conditioning and decoding, whereas the conditional prior $p_\eta(\mathbf{z}|y)$ is only needed to generate latent samples at inference time.

Similarly, unconditional generation is not the target setting of this paper because OD calibration is posed as conditional posterior estimation given observed traffic counts. Nevertheless, posterior diagnostics remain meaningful: for a fixed observation $y_o$, they reveal whether the learned model places probability mass near the target OD while retaining multiple plausible calibrated solutions. This is the relevant notion of uncertainty for the underdetermined calibration setting studied here.

### F.2 Evaluation Protocol and Budget Accounting

For neural posterior models, the reported RMSN is computed by drawing ten OD samples from the learned posterior, simulating the corresponding traffic counts, averaging the ten count vectors into one deterministic prediction, and then comparing that prediction with the ground-truth traffic counts. For SPSA and PC-SPSA, we report mean±std over ten converged runs.

We distinguish expensive SUMO calls from cheap analytical/noise-based augmentation. The full Ours model uses 150 SUMO calls versus 200 for SPSA, while Ours_small uses 10 expensive SUMO calls plus 140 augmented samples. Accordingly, the low-budget appendix table should be read as evidence about simulator-call efficiency under constrained budgets, not as a claim that every training input has the same cost.

The wall-clock interpretation follows the same distinction. Simulator-call plots isolate sample efficiency, whereas runtime depends on parallelism: under the hardware described in Appendix E, 200 SPSA calls correspond to roughly 40 simulator-hours, while 150 ControlVAE calls can be collected in approximately 48 minutes of simulator time under 40-way parallelism, excluding neural-network training overhead.

Finally, the Munich results should be interpreted as benchmarked calibration-recovery experiments built from synthetically perturbed historical OD demand, and the diffusion comparison should be interpreted as an off-the-shelf low-data baseline rather than a definitive statement about diffusion models for OD calibration.

The fixed-budget results and the low-budget appendix table answer different questions. Table 1 compares calibration quality at the budgets used for the main experiments, where the gap between CVAE-catt and ControlVAE can be modest in some settings. Table 3, by contrast, isolates the regime in which the analytical branch is most valuable: when expensive simulator calls are severely constrained. We therefore interpret the appendix low-budget analysis as the main evidence that the physics-aware branch improves stability and simulator-call efficiency.

Regarding empirical scope, ASNPE remains relevant related work, but we do not add a new ASNPE implementation in this revision. Accordingly, the revised manuscript limits its empirical claims to the evaluated transportation and neural baselines, while treating recent SBI approaches as complementary comparators in the related-work discussion.

### F.3 Controllability Parameters and Posterior Interpretation

The controllability parameter $\alpha$ determines how strongly the analytical branch contributes to the conditioning signal $y$: $\alpha = 0$ corresponds to purely simulator-conditioned flows, while larger $\alpha$ increases the influence of analytical guidance. In practice, we select $\alpha$ by grid search because it acts as a user-facing control on how much analytical structure is injected into training and inference.

The alignment weight $\gamma$ controls how strongly the analytical branch is matched to simulator outputs through $L_{\mathrm{MSE}}$. Together, $\alpha$ and $\gamma$ should be interpreted as the main controllability and sensitivity parameters of ControlVAE: $\alpha$ changes the condition seen by the CVAE, whereas $\gamma$ changes how strongly the analytical branch is encouraged to track the simulator.

The appendix distribution plots provide qualitative evidence that the learned conditional posterior both concentrates near the target OD and retains multiple plausible solutions under the same observed traffic pattern. This is the uncertainty notion relevant to the underdetermined calibration setting considered in the paper.

More broadly, these clarifications should be read together with the limitation section: the present study uses synthetic perturbation benchmarks on realistic traffic simulators, and further real-data studies will need to address detector coverage, route-choice mismatch, and temporal drift.

Relative to prior physics-informed VAEs, the analytical model in ControlVAE is not used as a hard governing-equation constraint on the decoded OD matrix. Instead, it acts as cheap differentiable guidance inside a conditional posterior estimator for simulator-based calibration. Relative to prior generative OD calibration work, the contribution is therefore the combination of a conditional generative posterior, controllable cross-attention fusion, and analytical guidance within one calibration pipeline.

We intentionally keep the main-paper discussion of posterior analysis lightweight and place the more interpretive discussion here because the current evidence is qualitative rather than a full uncertainty-calibration benchmark. The appendix plots should therefore be read as supporting diagnostics for posterior concentration and diversity, not as a claim that uncertainty has been exhaustively characterized in all deployment settings.

# G  Detailed Analysis

## G.1  Controllability of Physics Awareness

Table 3: Low-budget analysis of physics guidance. "SUMO calls" counts only expensive SUMO evaluations; Ours_small additionally uses 140 analytical/noise-augmented samples during fine-tuning.

| Method | SUMO calls | Munich 5-6 | Munich 6-7 | Munich 7-8 | Munich 8-9 | Munich 9-10 |
|---|---|---|---|---|---|---|
| SPSA | 200 | 18.00 (1.12) | 43.10 (0.24) | 55.89 (2.31) | 50.05 (0.61) | 36.13 (0.43) |
| CVAE-catt | 150 | 16.43 (0.79) | 30.72 (1.76) | 18.47 (1.75) | 21.60 (1.55) | 19.42 (1.15) |
| Ours_small | 10 | 15.44 (0.01) | 23.18 (0.32) | 18.33 (0.01) | 21.22 (0.46) | 18.45 (0.01) |
| Ours | 150 | 14.89 (0.56) | 21.74 (1.59) | 18.32 (1.83) | 21.02 (1.84) | 16.38 (1.02) |

The motivation of predicting $y^{Ana}$ is twofold. First, by introducing a trainable function that can approximate the simulator output to a certain extent, we aim to reduce the reliance on expensive simulator runs during the calibration process. The function can serve as a computationally efficient surrogate model. It primarily enhances computational efficiency, which is of particular importance when we have limited simulation runs. Second, it provides a consistent and informative context throughout the VAE architecture, enhancing the learning of meaningful relationships between the OD matrix and the traffic measurements. By combining the strengths of the simulator and the analytical equation, our framework aims to achieve accurate and computationally efficient OD calibration. As a surrogate model, the physics model itself doesn't directly improve performance. Our introduced control parameters and cross-attention mechanism, however, allow our underlying model to make use of relevant dynamics captured by the physics model.

To evaluate the controllability of physics awareness, we conducted a comparative analysis of four methods for estimating OD matrices in traffic flow modeling. Our innovative approach, Ours_small, demonstrates remarkable efficiency through a hybrid data strategy: we executed the simulator only 10 times and augmented this data by adding noise to create 140 additional samples without traffic flow data. The model was initially trained on the 10 complete samples, then fine-tuned with the 140 additional samples using $y = y^{Ana}$ in Equation 5. This appendix table is the basis for our low-budget sample-efficiency claim. Relative to SPSA's 200 expensive SUMO calls, Ours_small uses 10 expensive SUMO calls, which is a 95% reduction in simulator calls, plus 140 cheap analytical/noise-based augmented samples. By contrast, the full Ours model uses 150 SUMO calls versus 200 for SPSA, corresponding to a 25% reduction. We therefore revise the manuscript to report the exact budgets rather than the previous global "75% fewer simulations" wording.

As shown in Table 3, this approach yields impressive results across different time intervals in Munich. Despite using significantly fewer simulation runs (10) compared to other methods (150-200), Ours_small achieves competitive performance. For instance, in Munich 6-7, Ours_small achieves 23.18 ($\pm$0.32), while CVAE-catt, using 150 runs, reaches 30.72 ($\pm$1.76). Both our methods demonstrate superior stability with consistently lower standard deviations compared to SPSA, which shows high variability (ranging from 18.00 to 55.89) despite using 200 simulation runs. This supports a narrower interpretation of the physics component: it is most valuable as a differentiable inductive bias for stability and extreme-budget efficiency, not as evidence that the analytical model alone should yield uniformly large gains in every fixed-budget comparison.

## G.2  Convergence speed

We compare the convergence curves for the best RMSE achieved by SPSA, PC-SPSA, and our proposed ControlVAE method. While SPSA and PC-SPSA allow running two simulators concurrently per iteration, our approach enables up to 2n simulators during data collection, where n is the number of SUMO programs determined by CPU cores. This means the time for ControlVAE to accumulate 160 training points equals only 80/n iterations for SPSA and PC-SPSA. For fair comparison, we plot the results against the number of simulator executions rather than wall-clock runtime. During data collection, no optimization is applied to the generated data. We incrementally increase the training volume in units of 10 runs and record the distribution results. All three curves start from the 0th iteration. For display, SPSA and PC-SPSA are plotted from the 1st iteration, while ControlVAE is plotted from the 10th SUMO run. As shown in Figure

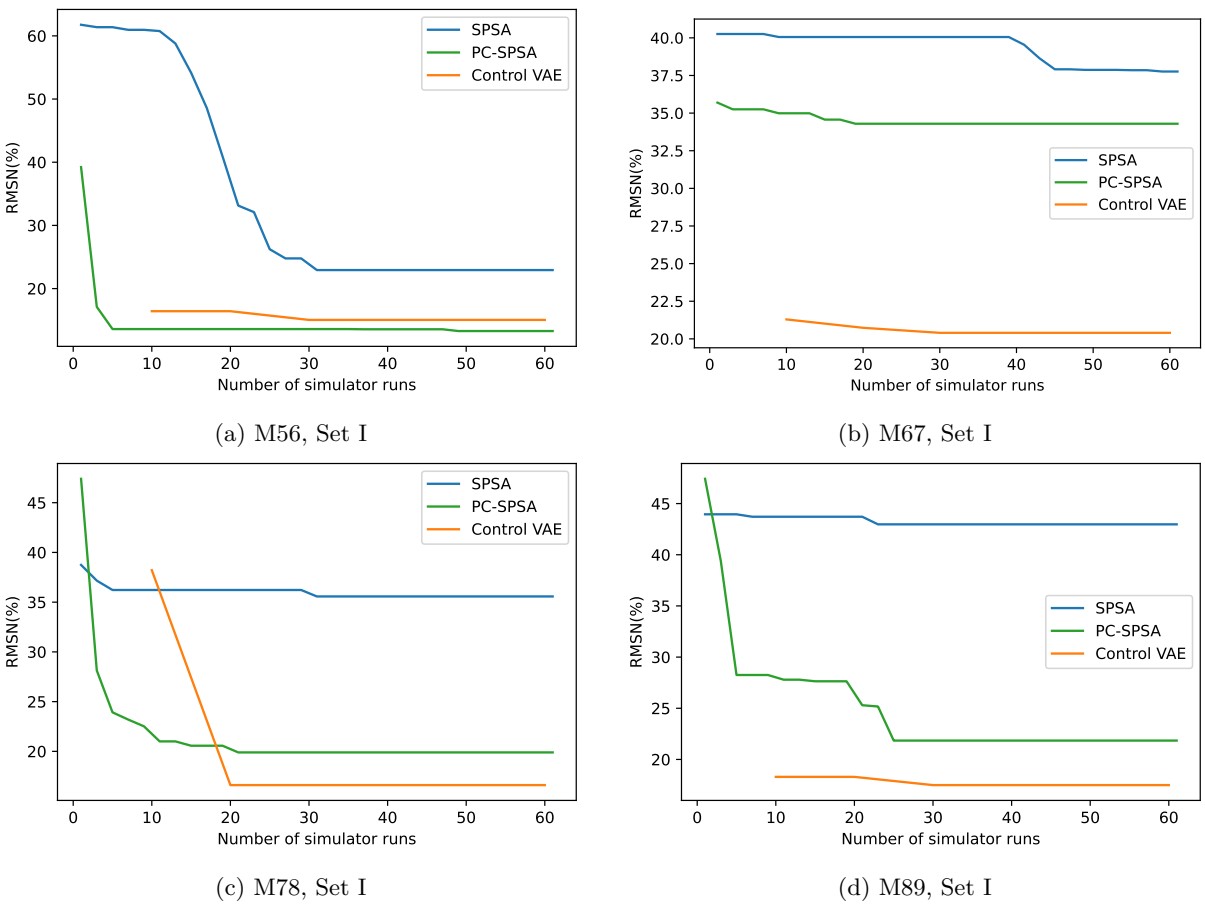

Figure 6: The converge curve based on the count of simulator running

6, PC-SPSA converges quickly but only exceeds ControlVAE for Setting M56, while failing to converge well on M67. In contrast, ControlVAE rapidly converges once the target result is obtained using just 10 SUMO runs. Considering the parallelizable data generation, this demonstrates the data efficiency of ControlVAE. By leveraging both simulation and analytical models with attention mechanisms, ControlVAE achieves accurate OD matrix calibration with fewer required simulator evaluations.

## G.3 Detailed Analysis with Potential Improvement and Diffusion Models

We conduct comprehensive evaluations by comparing our proposed method against the potential improvement of traditional approaches, i.e., CVAE-initiated SPSA and contemporary deep learning models, demonstrating its consistent superiority across all time intervals (Table 2). Traditional SPSA exhibits inconsistent performance, but its extension with CVAE initialization shows notable improvements, although it still lags behind our proposed method. This underscores two key insights: (1) SPSA's performance heavily depends on initialization quality, and (2) CVAE effectively generates meaningful initial OD values closer to optimal solutions.

Among modern deep learning approaches, the Transformer model demonstrates competitive and stable performance, while diffusion models underperform in our current setup. We emphasize that this comparison should be interpreted as an off-the-shelf diffusion baseline under a low-data OD-calibration protocol. It does not rule out the possibility that a diffusion model adapted specifically to this low-data regime could perform substantially better. In the current benchmark, we attribute the weak performance to a combination of limited training data, structural difficulty in modeling OD dependencies, sensitivity to initialization, and mismatch between the standard diffusion noise process and the observed OD distribution.

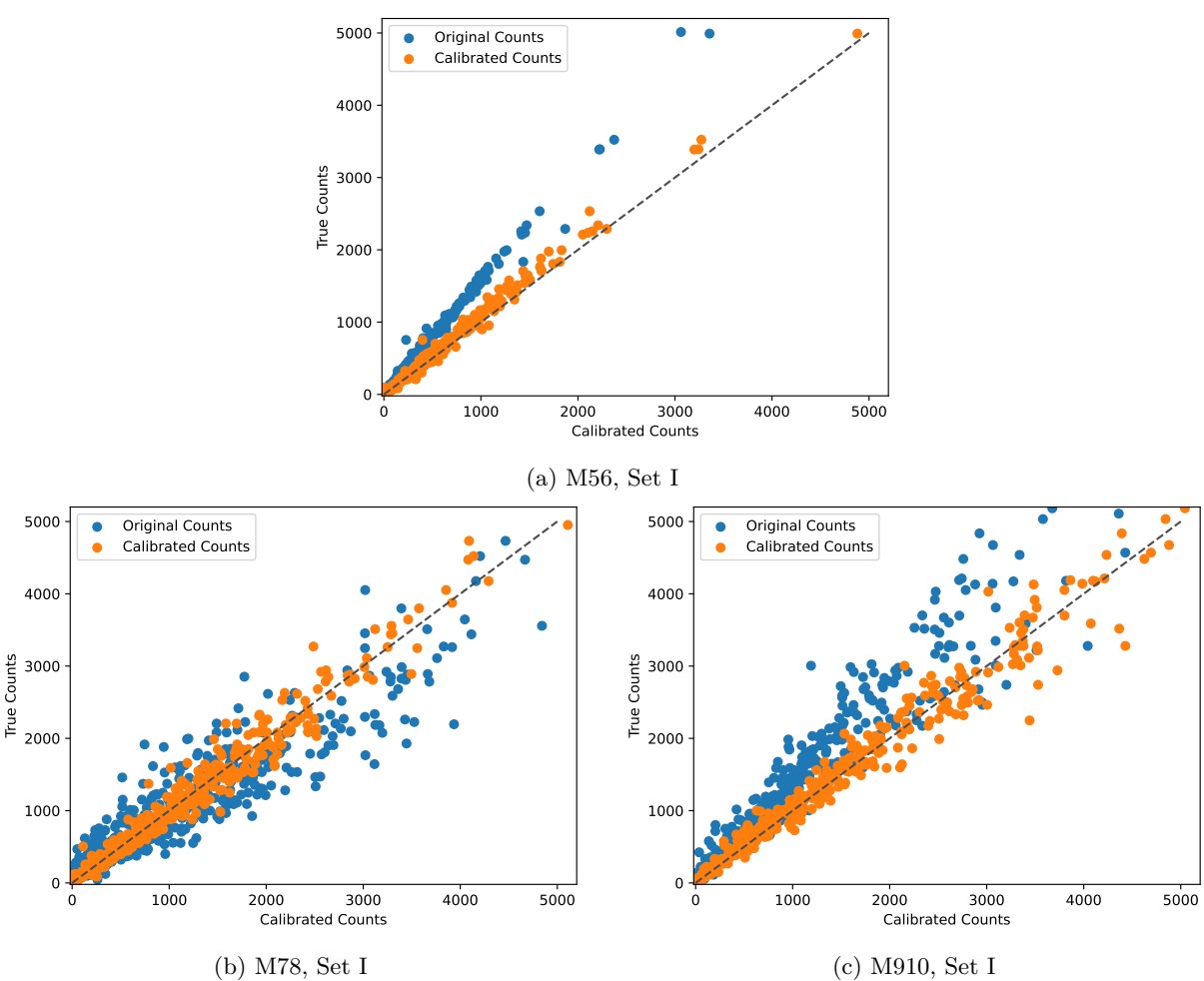

Figure 7: Extra Calibration Results on Traffic Counts.

## H Limitation

Despite the limited focus on employing deep generative models for the complex task of OD calibration, our study presents an innovative application of Variational Autoencoders (VAEs) to model OD distribution from a Bayesian perspective. We explore a cultured analytical approximation of traffic network mechanisms, integrating it within the VAE framework using a controlled cross-attention module. Our current diffusion comparison should be read as an off-the-shelf low-data baseline rather than a definitive negative result for diffusion models in OD calibration. Diffusion models remain a promising future direction, especially if adapted to the low-data and non-Gaussian characteristics of this task.

Our study faces another limitation stemming from the reliance on synthetic data derived from observed OD demand, potentially resulting in a significant dependence on the SUMO simulator for the relationship between OD demand and traffic data. Although acquiring real data is challenging and may be restricted for scientific use due to data protocols, we maintain that utilizing real data in future work is crucial. Real-world scenarios encompass a diverse array of weather and human factors, making them more complex than simulations and providing more valuable insights. Furthermore, awareness of the observed OD demand enables the evaluation of differences between calibrated OD values and actual values, as well as the estimation of initial OD distribution proximity to the real OD distribution when constructing scenarios. This is important because a reasonable initial demand distribution may be a prerequisite for recovering the true distribution. In real traffic data, additional issues such as missing detectors, route-choice mismatch, and temporal drift

can further complicate calibration. In summary, although synthetic data facilitates initial methodological development, real-world data will be required to fully validate our approach and apply it in practice. Ongoing work is focused on gaining access to real transportation data under appropriate privacy agreements.

While substantial progress has been made in recent years, current models, including Control VAE, have not yet reached the desired level of accuracy for urban travel demand calibration using deep generative models. Nevertheless, we argue that the margin of error is swiftly decreasing as researchers enhance their understanding of and address potential risks associated with urban travel demand calibration. In particular, illogical or unreasonable generated samples could have detrimental consequences for urban development. Given these considerations, we recommend that researchers adopt a human-centric approach to develop a responsible traffic system, underpinned by neural networks that embody causality, fairness, interpretability, privacy, security, and accountability.

