# OpenReview forum: "Physics-Aware Variational Autoencoder for Urban Travel Demand Calibration"
_TMLR — Decision pending for TMLR_

### Review · Reviewer_L31x · 2025-12-11

**Summary Of Contributions:**

The paper addresses origin–destination (OD) demand calibration for large-scale urban traffic simulators. It proposes ControlVAE, a conditional VAE that incorporates a physics-based analytical traffic meta-model via cross-attention. A scalar parameter α interpolates between flows predicted by the analytical model and those generated by the simulator, allowing a trade-off between physics guidance and purely data-driven behavior. Experiments on the Sioux Falls network and a large Munich network show improved fit to traffic counts (RMSN) and reduced simulator calls compared to SPSA/PC-SPSA and several neural baselines (CVAE variants, transformers, diffusion models).

**Audience:**

Yes

**Audience Explanation:**

Traffic simulator is of high importance of autonomous driving.

**Broader Impact Concerns:**

No.

**Claims And Evidence:**

Yes

**Claims Explanation:**

1. Relevance: OD calibration for metropolitan-scale digital twins is practically important, and reducing the number of expensive simulator runs tackles a real bottleneck.
2. Method: Combining a CVAE with a linear physics meta-model using cross-attention is intuitive and well-motivated. The control parameter α is a clear and useful design choice.
3. Empirics: The paper evaluates on both a standard benchmark and a realistic large network, with consistent gains in RMSN and clear evidence of better sample efficiency than strong optimization-based baselines.
4. Clarity of high-level idea: The overall architecture and motivation (posterior over OD matrices to handle underdetermination) are well explained at a conceptual level.

**Requested Changes:**

- Clean up writing, remove TODOs and duplicated text, and consolidate the physics meta-model description with consistent notation.
- Clarify what is novel relative to existing physics-informed VAEs and prior generative OD calibration work.
- Add at least some additional analysis of the learned posterior (or uncertainty) and sensitivity to key hyperparameters (especially α and the physics-alignment weight).
- If feasible, include a small real-data example or a more detailed discussion of issues that would arise when moving from synthetic to real traffic data.

---

### Review · Reviewer_9wRQ · 2025-12-18

**Summary Of Contributions:**

This paper proposes ControlVAE, which incorporates a differentiable physics-based traffic model for OD matrix calibration. The model provides cheap approximate gradients, reducing dependence on expensive simulator calls.

**Audience:**

Yes

**Audience Explanation:**

The core idea of injecting a cheap differentiable physics surrogate to reduce simulator calls is relevant, but the physics model here is very simple (static linear assignment) and its contribution is marginal per their own ablations.

**Claims And Evidence:**

No

**Claims Explanation:**

- The paper claims 75% fewer simulation, but in Table 3, SPSA has 200 runs and ControlVAE has 150 runs. Ours_small is a secondary variant that still requires generating 140 augmented samples, which isn't free. I am confused why this is 75% fewer.

- "Up to 40% RMSN reduction" appears to be an overstatement. In Table 1, only one column shows significant improvement while others show modest or no improvement. Also why is PC-SPSA is missing for Sioux Falls?

- The main results only compares to SPSA. Are there other methods to compare with? What about ASNPE which was cited in the paper? Without extensive empirical evaluation, it is very hard to validate its contribution.

- According to the Appendix, the munich dataset starts with some OD estimate $\hat{d}$, then perturb it to create ground truth x_c. This experiment relies on synthetic data and hasn't been tested on real data. Additionally, Munich at different hours is the same network topology and this alone may not single-handedly provide evidence for the claim. The Sioux Falls dataset looks like a toy dataset with 9 OD zones and 39 sensors, which doesn't test high-dimensional performance. By ML standards, this is like only evaluating on one dataset, and is really lacking in terms of empiricism.

- Table 2 shows the diffusion model underperforms, but it may be unfair, as there was no attempt to make adapt diffusion models to the low-data regime.

- The ablation shows a very small gap between CVAE-catt and controlVAE, eg. 22.41 vs 22.02 Munich 6-7, 16.34 vs 16.28 Munich 9-10. This paper's main contribution is physics-informed VAE, but it doe not seem like the physics component is contributing much. If removed, what is the physics actually contributing?

**Requested Changes:**

- Clarify the "75% fewer simulations" claims. Table 3 uses 150 vs 200 runs for example - explain the calculations behind or revise the claim.

- Additional baselines - is ASNPE applicable or are there others?

- The method outputs a distribution, but Table 1 reports a single RMSN value. Explicitly explain how this was obtained.

- Justify the physics component's contribution. The performance difference is too small. Provide additional analysis showing why it helps or temper claims about it being a main contribution.

- Adapt the diffusion model to a low-data resume or clearly frame this as showing that off-the-shelf diffusion omdels are unsuitable for this task rather than using it as a baseline comparison. Currently, it seems there is no effort put into it and that it is designed to fail.

- discuss wall-clock time comparison in addition to simulation count comparisons. Figure 3f looks confusing to me because SPSA is sequential but your method can be run in parallel - comparising the two in a loss vs runs plot isn't exactly apple-to-apple.

- minor issue: the "TODO" comment in Appendix C. Typos "ariational encoders" page 2 and "disreibution" page 1.

- minor: specify the encoder/decoder architecture such as number of layers, hidden dimensions etc.

---

### Review · Reviewer_Yx4Z · 2026-03-26

**Summary Of Contributions:**

The paper focuses on a traffic origin-destination (OD) calibration problem. Specifically, it aims to determine an underdetermined OD matrix (e.g., of size $M \times M$) given $n$ traffic observations from different road segments.

To approach this problem, the authors propose a conditional variational autoencoder (CVAE) framework. Taking the OD demand matrix $D$ as input, the encoder $\phi$ learns a latent variable $Z$ to approximate the posterior distribution $q_\phi(Z|D, Y)$, where $Y$ represents traffic flow observations. These observations are formulated as a convex combination of simulated traffic flow and flow generated by an analytical equation. Furthermore, because $D$ and $Y$ originate from different modalities, a cross-attention mechanism is applied within the encoder to generate $Z$ given $D$ and $Y$.

In the decoding phase, a decoder $\psi$ outputs the OD prediction $D$ following the distribution $p_\psi(D|Z, Y)$, conditioned on both the latent variable $Z$ and the observation $Y$. The model is trained using the classical variational lower bound of the CVAE, augmented with a mean squared error component for the reconstructed traffic flow $Y$ derived from the analytical equations and the simulator.

During inference, given $Y$, the latent variable $Z$ is generated using a surrogate network $p(Z|Y)$, which acts as a marginal distribution conditioned solely on $Y$. And then decode the desired $D$ by the learned Decoder. This surrogate distribution is jointly learned during the training process, where $Z$ is first sampled from $p(Z|Y)$ before computing $q_\phi(Z|D, Y)$, if I’m understanding correctly since this part is not described very clearly in the paper.

Finally, the authors validate their approach through quantitative experiments on the Munich regional network and the Sioux Falls network, benchmarking performance against traditional methods, neural baselines, and recent deep learning-based generation techniques.

**Additional Comments:**

I would recommend a major revision based on my comments above.

**Audience:**

Yes

**Audience Explanation:**

An application study of CVAE model to the OD calibration problem.

**Broader Impact Concerns:**

No concerns.

**Claims And Evidence:**

No

**Claims Explanation:**

**Uncertainty Modeling:** In the abstract and introduction, the authors highlight that a key contribution is the model's ability to capture uncertainty as a distribution rather than a point estimation. However, the paper lacks sufficient qualitative and quantitative analysis to demonstrate this. Additionally, are the learned distributions multimodal or restricted to a single mode?


**Unconditional Generation:** Could the authors provide a qualitative study of unconditional generation? Specifically, it would be helpful to see the generation of the OD matrix $D$ directly from the latent variable $Z$, without the condition $Y$. This would help verify if the latent space has successfully learned a representative prior of valid OD configurations, rather than simply relying on the conditioning observations.


**Training Mechanics:** How exactly does the surrogate distribution $p(Z|Y)$ interact with the $Y$ output from the analytical equations and the simulator during training?


**Differentiability:** What does the reconstructed traffic flow $Y$ error in the training objective refer to? Specifically, are the analytical equation and the simulator learnable or frozen? Are they neural network-based and differentiable to allow backpropagation?


**Matrix P:** Regarding Equation 3 in Section 3.3, how is the matrix $P$ learned during training, and how is it utilized during inference?


**Ablation Studies:** The cross-attention fusion module may require an ablation study to properly justify its effectiveness and necessity in the architecture.


**Independence Assumptions:** On page 6, just before Equation 5, the authors state that the posterior $q_\phi(Z|D, Y)$ is modeled as a Gaussian distribution with a diagonal covariance matrix. This implies that different entries in the OD matrix are assumed to be strictly independent. Although I am not a domain expert in OD calibration, from an outside perspective, it intuitively seems that different OD pairs are not independent. This seems like a very strong—and potentially unrealistic—assumption for this specific problem.


**Experimental Baselines:** In Section 4.4, all comparison methods are from 2024 or earlier. Are there newer state-of-the-art methods from 2025 that the authors can include to ensure the benchmarking is up to date?


**Inference Clarity:** The explanation of the inference stage only appears on page 7 (Section 3.4), and not very clearly. Given that OD calibration is the core problem, the inference process should be outlined much earlier (e.g., in the introduction) to avoid confusion.

**Requested Changes:**

**Typo:** In the first paragraph of the introduction, "distribution" is misspelled in the context of estimating the OD matrix distribution.


**Citations:** The Related Work section lacks literature from 2025. This should be updated to include the most recent progress in the field.


**Figure 2:** The $p(Z|Y)$ surrogate network is missing from Figure 2. Since it is a key component for the loss function and inference, it should be included.


**Notation Consistency (Parameters):** In Figure 2, $\psi$ is assigned to the decoder $p_\psi(D|Z, Y)$. However, in Equation 6 of Section 3.5, the same $\psi$ is used for the distribution $p_\psi(Z|Y)$. If I’m understanding correctly, these are distinct components of the architecture; using the same parameter suggests weight sharing, which is likely a notation error that needs to be corrected for clarity.


**Notation Consistency (Variables):** Equations 1 and 2 use $H$ for the latent representation, while Figure 2 uses $Z$. Please ensure consistent labeling of the latent space.


**Completeness of study**: there are many “TODO” explicitly stated in appendix. E.g Appendix C and D, which are very confusing to readers.

---

### Decision · Action_Editor_RJCT · 2026-05-29

**Recommendation:** Accept as is

**Audience:**

Yes

**Audience Explanation:**

Yes. The paper addresses an important and practically relevant problem in large-scale urban mobility modeling and digital twins, combining machine learning, generative modeling, and physics-informed learning. The proposed framework and findings should be of interest to researchers working on scientific machine learning, simulation-based inference, transportation AI, and physics-guided generative models.

**Claims And Evidence:**

Yes

**Claims Explanation:**

The paper presents a technically sound and well-motivated framework for sample-efficient OD calibration by integrating differentiable physics guidance into a conditional VAE architecture. The claims are generally supported by empirical evaluations on both benchmark and metropolitan-scale traffic networks, and the authors have adequately clarified the methodology, experimental setup, and limitations through the revision process.